# Dual Impact of a Benzimidazole Resistant β-Tubulin on Microtubule Behavior in Fission Yeast

**DOI:** 10.3390/cells10051042

**Published:** 2021-04-28

**Authors:** Mamika Minagawa, Minamo Shirato, Mika Toya, Masamitsu Sato

**Affiliations:** 1Laboratory of Cytoskeletal Logistics, Department of Life Science and Medical Bioscience, Graduate School of Advanced Science and Engineering, Waseda University, 2-2 Wakamatsucho, Shinjuku-ku, Tokyo 162-8480, Japan; m-minagawa852@akane.waseda.jp (M.M.); mina.wd-yz@moegi.waseda.jp (M.S.); mikatoya@aoni.waseda.jp (M.T.); 2Faculty of Science and Engineering, Global Center for Science and Engineering, Waseda University, 3-4-1 Okubo, Shinjuku-ku, Tokyo 169-8555, Japan; 3Institute for Advanced Research of Biosystem Dynamics, Waseda Research Institute for Science and Engineering, Graduate School of Advanced Science and Engineering, Waseda University, 3-4-1 Okubo, Shinjuku-ku, Tokyo 169-8555, Japan; 4Institute for Medical-Oriented Structural Biology, Waseda University, 2-2 Wakamatsucho, Shinjuku-ku, Tokyo 162-8480, Japan

**Keywords:** tubulin, microtubules, benzimidazole, fungicides, cytoskeleton, fission yeast

## Abstract

The cytoskeleton microtubule consists of polymerized αβ-tubulin dimers and plays essential roles in many cellular events. Reagents that inhibit microtubule behaviors have been developed as antifungal, antiparasitic, and anticancer drugs. Benzimidazole compounds, including thiabendazole (TBZ), carbendazim (MBC), and nocodazole, are prevailing microtubule poisons that target β-tubulin and inhibit microtubule polymerization. The molecular basis, however, as to how the drug acts on β-tubulin remains controversial. Here, we characterize the *S. pombe* β-tubulin mutant *nda3-TB101*, which was previously isolated as a mutant resistance to benzimidazole. The mutation site tyrosine at position 50 is located in the interface of two lateral β-tubulin proteins and at the gate of a putative binging pocket for benzimidazole. Our observation revealed two properties of the mutant tubulin. First, the dynamics of cellular microtubules comprising the mutant β-tubulin were stabilized in the absence of benzimidazole. Second, the mutant protein reduced the affinity to benzimidazole in vitro. We therefore conclude that the mutant β-tubulin Nda3-TB101 exerts a dual effect on microtubule behaviors: the mutant β-tubulin stabilizes microtubules and is insensitive to benzimidazole drugs. This notion fine-tunes the current elusive molecular model regarding binding of benzimidazole to β-tubulin.

## 1. Introduction

Microtubules are a cytoskeleton essential for a number of cellular events in eukaryotes and in prokaryotes. In eukaryotes, microtubules are organized in the cytoplasm to form a network array during interphase. Upon entry into mitosis, cytoplasmic microtubules are reorganized into mitotic spindle. Microtubules consist of tubulin dimers, each of which is made of two structurally similar proteins, α-tubulin and β-tubulin [1,2]. Microtubules can be polymerized by the addition of αβ-tubulin dimers on the top of the microtubule bundle, and also depolymerized by removal of the dimers off the end of the bundle [3]. As αβ-tubulin dimers are aligned in a unidirectional manner, a microtubule filament possesses the polarity in direction: the end that exposes α-tubulin is called the minus end, whilst the other end exposing β-tubulin is called the plus end. Tubulins are a GTP-binding protein (GTPase), and they are GTP-bound before and right after incorporation into a microtubule, whereas GTP is hydrolyzed into GDP gradually after incorporation. Microtubules are stable in the GTP-bound form but unstable when hydrolyzed into GDP. These are the properties of tubulin dimers in polarity, and in nucleotide binding conditions, a microtubule displays dynamic instability in which the plus end is more dynamic than the minus end.

In cells, a number of microtubule-associated proteins (MAPs) modulate the behavior of microtubules to function in each cellular event [4]. Some MAPs are known to localize to the plus end of a microtubule bundle (+TIPs). The γ-tubulin and its regulatory factors (the γ-tubulin complex) localize to the minus end. The end-binding protein EB1 (Mal3 in fission yeast), a +TIP, is known to bind to the plus ends of growing microtubules in addition to the lateral surface [5,6].

Polymerization and depolymerization are also promoted by MAPs in cells. Conserved MAP family proteins TOG (tumor overexpressed gene)/Dis1/XMAP215 serve as processive polymerase in eukaryotic cells [7]. Fission yeast has two paralogues of the TOG family, Dis1 and Alp14. Mutant cells lacking Dis1 (*dis1*∆) display an almost normal array of cytoplasmic microtubules in interphase, whereas *alp14*∆ cells display short and fragile microtubules [8,9,10], indicating that Alp14 is the major microtubule polymerase in cells [11].

On the other hand, some subfamilies of the kinesin superfamily, such as kinesin-13 and kinesin-8, are shown to function as microtubule depolymerase [12,13,14]. Fission yeast does not appear to have kinesin-13 family members that contain a conserved kinesin domain in the middle of the gene. Instead, there are two kinesin-8 proteins, Klp5 and Klp6, which function as a heterodimer in cells [15,16,17]. Deletion of either kinesin-8 gene results in an extension of cytoplasmic microtubules that do not stop growing at cell tips, rather they continue to grow and show curvy bundles. This demonstrates that those kinesin-8s are factors promoting microtubule depolymerization.

In addition to the native properties of αβ-tubulin dimers and microtubules, those MAPs contribute to generate precise and dynamic properties of microtubules so that the cytoskeleton plays pivotal roles in many aspects of cellular events.

A large number of microtubule drugs that affect the dynamics of microtubules have been developed. Particularly for agricultural purposes, antifungal agents (fungicides) have been introduced and widely used since the 1970s.

Benzimidazole compounds are representative antifungal and antiparasitic agents, which include prevailing derivatives such as MBC (methyl-2-benzimidazole carbamate, also known as carbendazim [CBZ]), TBZ (thiabendazole), benomyl, and nocodazole [18]. In general, those compounds inhibit the growth of fungi by blocking mitotic divisions, most likely through inhibition of spindle assembly. Antimitotic microtubule drugs are also introduced as anticancer agents. These include colchicine, paclitaxel (taxol), and vinca alkaloids such as vincristine and vinblastine [19].

Reflecting those various demands, a large number of studies have been performed to understand how the drugs affect microtubule behaviors. Another important strategy to understand the molecular and pharmacological mechanisms is the genetic approach; for instance, isolation of mutants that are either supersensitive or resistant to these drugs.

Mutants supersensitive to microtubule drugs could be brought on by mutations in various genes encoding MAPs. By contrast, most of mutants resistant to the drugs isolated either naturally or artificially contained mutations in β-tubulin genes, and analyses for the mutation sites contributed to the identification of the putative sites where the drugs bind. In many fungal species, β-tubulin mutants have been isolated that show resistance to benzimidazole compounds.

Those include mutations at A165 (N165 in mammals), H6, E198, or F200 of the filamentous fungus *Aspergillus nidulans* [20,21]; at E198 of the red bread mold *Neurospora crassa* [22] and the plant pathogen *Podosphaera xanthii* [23]; at E198 or F200 of the necrotrophic fungus *Botrytis cinerea* [24]; various mutations at 40–46, 123, 194–198, 282–297, 318–320 of the budding yeast *Saccharomyces cerevisiae* [25]; and at E198, F200 or L240 of the cereal eyespot fungi *Tapesia yallundae* and *Tapesia acuformis* [26]. The β-tubulin mutants in F167 or F200, conferring resistance to benzimidazole in parasitic nematodes such as Trichostrongylid species including *Haemonchus contortus*, have been also isolated [27,28].

In this study, we analyze a benzimidazole-resistant mutant of the fission yeast *Schizosaccharomyces pombe*. As microtubules are similarly organized in mammalian and *S. pombe* cells, fission yeast is widely used as a model organism to understand regulations of microtubules in living cells.

*S. pombe* has two α-tubulin proteins and a β-tubulin protein. One of two α-tubulin proteins, Nda2 (also known as Atb1), is essential for growth, whereas the other one, Atb2, is dispensable for growth [29,30]. The only β-tubulin protein Nda3 (also known as Alp12) is essential for growth [31].

Applying the inhibitory effect of benzimidazole compounds on the growth of *S. pombe* cells, a genetic screen was performed to isolate mutants that exhibited altered sensitivity to the drugs. The *ben1* mutant was isolated as a mutant that confers resistance to TBZ, and the responsible gene was the β-tubulin gene *nda3* [32,33]. From analogy to the E198A mutant of *P. xanthii*, the E198A mutation was recently introduced to the *S. pombe* β-tubulin and found to similarly confer resistance to MBC on agar plates [34], although no analyses were made for microtubule behaviors in the mutant.

## 2. Materials and Methods

### 2.1. Yeast Strains and Genetics

*S. pombe* strains used in this study are listed in Appendix A. Standard media and methods were used for *S. pombe* genetics [35]. Briefly, YE5S was used as rich medium throughout the study, and SPA (sporulation agar) was used for genetic crossing to induce meiosis and sporulation. EMM (Edinburgh minimal medium) supplemented with nitrogen (ammonium chloride) and carbon (D-glucose) sources as well as amino acid supplements was used for observation under the microscope.

For construction of gene knock-out and knock-in strains, PCR-based conventional methods were used [36,37,38].

### 2.2. Insertion of the Blasticidin S Resistance Gene

To construct the *nda3*^+^ and *nda3-TB101* strains with the *bsd* marker gene, the gene conferring blasticidin S resistance was inserted at the downstream of the *sdh8*^+^ gene, which located next to the *nda3*^+^ gene locus, as illustrated in Figure 1B. This was to avoid modification of the promoter and terminator regions of the *nda3-TB101* gene.

To minimize effects of marker insertion on expression of the *sdh8* gene, we inserted the *adh1* terminator for *sdh8* in front of the *bsd* gene. For easy construction, we used the plasmid pFA6a-3HA-bsd, which was modified from pFA6a-3HA-kan [36]. A DNA fragment containing the gene encoding three copies of HA (3HA epitopes), the *adh1* terminator, and the *bsd* marker gene was PCR-amplified and inserted at the right after the termination codon of the *sdh8* gene so that the HA tag could not be translated.

The allele of the resultant strain was termed *nda3-TB101-bsd*, and the *bsd*-marked strains were used for most experiments presented in this study except in a part of Figure 1A and in Figures 4 and 5 where MDR-sup strains were used (Appendix A).

### 2.3. Prediction and Modelling of the Protein Structure

For prediction of the secondary and tertiary structures of wild-type and mutant proteins, amino acid sequences were subjected to the protein folding prediction program PHYRE2 (http://www.sbg.bio.ic.ac.uk/phyre2/, accessed on 12 March 2021) [39]. Amino acid sequences of *S. pombe* β-tubulin Nda3 (WT) and the Nda3-TB101 (Y50S substitution) were used as queries for prediction of structures, unless otherwise specified. The closest model, c2p4nB, was selected by the program as a template to predict structures of queried proteins. Alignment of mammalian αβ tubulin dimers (α, a hybrid model of human and bovine; β, bovine) and the N-terminal fragment of Mal3 shown in Appendix A is based on the model dataset 4ABO.pdb provided in the previous study [6]. The bovine αβ-tubulin dimer shown in Supplemental Appendix A is based on the model dataset 1TUB.pdb provided in a previous study [1].

### 2.4. Microscopy

Methods for preparation of cells under our live-cell imaging system has been described previously [38]. Briefly, cells were mounted to a 35 mm glass-bottom dish (Matsunami, Kishiwada, Osaka, Japan) precoated with soybean lectin (Sigma, St. Louis, MO, USA) and filled with the minimal medium EMM. Live-cell imaging was performed with the DeltaVision-SoftWoRx system (Applied Precision, Issaquah, WA, USA).

For visualization of microtubules, strains expressing the fusion gene of GFP and Atb2 (α2-tubulin) were used. The construct is called Z2-GFP-Atb2 [40]. Briefly, the fusion gene *GFP-atb2* placed under the *atb2* native promoter (Patb2) and the G418 resistance gene (Patb2-*GFP-atb2-kan*) were in tandem inserted at the Z2 locus of a chromosome. The resultant strain expresses GFP-Atb2 in addition to the endogenous Atb2.

For measurement of dynamics of cytoplasmic microtubules, behavior of the plus end of a bundle was chased in a time-lapse recording filmed every 10 s, and catastrophe was defined as an event in which a polymerizing microtubule turns into depolymerization, whereas rescue was as an event the other way around. Values are shown as mean ± S.D. of those from at least 40 microtubule bundles except in Figure 8A (20 bundles). Observation was done at 30 °C (Figures 3, 4B, 5 and 8), 36 °C (Figure 7B,C), or 25 °C (Figure 7D) in YE5S with or without microtubule poisons. For control, DMSO was used unless otherwise stated.

To quantify signal intensities of Mal3-GFP and Alp7-3GFP in cells (Appendix A), overnight cultures were prepared and then observed in EMM. Punctate GFP signals at the ends of microtubules were chosen as plus-end localizations for quantification, but other internal in-line localizations along microtubules were excluded. Quantification was done with ImageJ/Fiji [41]. Fluorescence intensities per pixel were normalized with cytoplasmic intensities as background and shown with arbitrary units in figures.

### 2.5. Drug Assays

Microtubule drugs, such as TBZ (thiabendazole; Sigma, T8904, St. Louis, MO, USA), MBC (methyl-2-benzimidazole carbamate/carbendazim; Sigma, 378674, St. Louis, MO, USA), and nocodazole (Noc; Sigma, M1404, St. Louis, MO, USA), were added to a solid or liquid YE5S medium at indicated concentrations in each figure legend. To test the sensitivity of *S. pombe* cells to nocodazole (Figures 4 and 5C), strains with the “marker-less” MDR-sup background were used [42].

### 2.6. Cold-Shock Assays

WT and *nda3-TB101* cells were grown in the YE5S medium at 25 °C until the mid-log phase and cultures were placed on ice for 30 min to induce microtubule depolymerization. The cultures were then returned to 25 °C to induce microtubule regrowth. Samples were taken every minute after the temperature shifted back. The samples for 0 min were taken during on ice just prior to induction of regrowth and the samples before cold shock were taken before putting them on ice. Samples were fixed in 30% methanol for 2 min before washing out with phosphate buffered saline (PBS). Samples were then mounted for microscopy.

### 2.7. In Vitro TBZ Binding Assays

For in vitro binding assays, cDNAs encoding *S. pombe nda3*^+^ and *nda3-TB101* were prepared and cloned into the pGEX-KG vector so that fusion proteins GST-Nda3 (WT) and GST-Nda3-TB101, respectively, could be expressed in the *E. coli* XL1-Blue cells. The vector was also used for expression of GST. Transformant *E. coli* cells were precultured in 500 mL 2xYT with ampicillin and expression was induced with 0.1 mM IPTG and incubated at 20 °C for 6 h. Cells were harvested by centrifuge and pellets were stored frozen at −80 °C overnight. Cells were suspended with a 10 mL PBS* buffer (1.5 mM NaH_2_PO_4_·2H_2_O, 8.2 mM Na_2_HPO_4_, 145 mM NaCl, 3% NP-40, 10 mM DTT, 1 mM PMSF, complete EDTA free (Roche, Mannheim, Germany, 1 tablet per 45mL PBS)) and lysed through sonication on ice for 7 cycles of 1 min procedure (giving pulses 30 times/min, with a 1 s interval after each pulse) using the sonicator VP-50 (TAITEC, Koshigaya, Saitama, Japan). Lysates were centrifuged for three times to remove cell debris. Subsequently, 300 µL of Glutathione Sepharose 4B beads (GE Healthcare, Uppsaka, Sweden) was added to the final supernatants and incubated at 4 °C for 1 h. The beads with GST-fusion proteins were collected via spin down. Before final suspension with PBS*, 2.5 µL (0.83%) of the precipitated beads was applied for SDS-PAGE and staining with Coomassie brilliant blue (Bio-rad, Hercules, CA, USA) to confirm binding of GST-Nda3 and GST-Nda3-TB101 to the sepharose. From the electropherogram, ~0.80 nmol of GST-fusion proteins and ~3.5 nmol of GST were estimated to reside in the column as a carrier for affinity absorption. The rest of the beads were suspended with 2 mL of PBS* to apply for a chromatography column (Bio-rad, Hercules, CA, USA). After washing with 3 mL of PBS* twice, 1.5 mL of 0.8 μM TBZ/PBS* was applied to the column. A flow through (fraction 0) and eluted samples (1–9) were sequentially collected (0.1 mL for each). Fractionated samples 1–9 were then subjected to an absorption spectrophotometer to measure the absorbance at 297 nm for TBZ [43].

### 2.8. Statistical Analysis

A Student’s *t*-test was used to evaluate the statistical significance of the difference between two values, except for analyses in Appendix A, in which Welch’s *t*-test was used. All data are presented as the mean ± standard deviation.

## 3. Results

### 3.1. The nda3-TB101 Mutant Is Resistant to TBZ and MBC

We first reconfirmed the resistance of the *nda3-TB101* mutant to TBZ in a plate-based assay. The wild-type (WT) strain was tolerant to TBZ at a lower concentration (10 μg/mL) but sensitive to higher doses of TBZ: no WT colonies were grown at 50 μg/mL TBZ (Figure 1A). The *nda3-TB101* mutant, on the other hand, survived even at 50 μg/mL TBZ, confirming that the mutant is resistant to the microtubule drug TBZ, as shown in previous studies [32,33].

For further experiments, the *bsd* gene, a selection marker that confers resistance to blasticidin S, was inserted near the *nda3* locus (for both alleles of *nda3*^+^ (WT) and *nda3-TB101* (mutant)). This is useful for selection of colonies containing the *nda3-TB101* mutation after genetic crossing, for example. In order to preserve the promoter and terminator regions of the *nda3* gene, the *bsd* marker was inserted not right after the *nda3* gene but after the neighbor gene *sdh8* (Figure 1B and Materials and Methods). The resultant strains are denoted as *nda3-TB101-bsd* and *nda3^+^-bsd.* In plate-based assays, the marked strains grew normally as the WT strain did, and the mutant allele *nda3-TB101-bsd* remained resistant to TBZ as the original mutant without the marker did (Figure 1A). These results verified that insertion of the gene cassette containing the *adh1* terminator and the *bsd* marker gene affected neither growth in the absence of drugs nor resistance to TBZ. The *nda3-TB101-bsd* strain was thus validated for use in further experiments.

### 3.2. Nda3-TB101 Has a Mutation Near the H1′-S2 Loop of β-Tubulin

We next analyzed the sequence of the *nda3-TB101* mutant gene and identified a single mutation in the β-tubulin coding sequence: a tyrosine residue (Y50) was substituted to serine (Y50S). As Figure 2A shows, Y50 is evolutionarily conserved in β-tubulin from yeast to human. Domains and sequences of α- and β-tubulin proteins have been extensively analyzed by a number of biochemical and structural studies. Y50 locates in the N-terminal half, which is called the nucleotide binding domain [1] (Figure 2B). The domain of β-tubulin for α-tubulin interaction is located in the intermediate domain [1], suggesting that Y50S mutation may not affect interaction between α- and β-tubulin in a heterodimer.

In line with this, the *nda3-TB101* mutant was viable without any apparent defects in vegetative growth (see Figure 1A), indicating that Y50S substitution did not severely deteriorate formation of αβ-tubulin dimers.

Figure 2C illustrates the structure of the β-tubulin Nda3 predicted by the program for protein folding prediction, PHYRE2 (http://www.sbg.bio.ic.ac.uk/phyre2/, accessed on 12 March 2021). The predicted entire structure of the mutant protein Nda3-TB101 was similar to that of wild-type Nda3 (top, Figure 2C), except for the N-terminal end, which is slightly bent by Y50S substitution (bottom). Thus, Y50S mutation may rather give a local impact on the protein, although we cannot exclude the possibility that Y50S brings conformational change around the region.

Y50 is predicted to locate juxtaposed to the H1′-S2 loop [Y51-R62] that is exposed on the surface of β-tubulin [44,45] (top, Figure 2C). The H1′-S2 loop and the neighboring H1 helix, as well as some other helices of β-tubulin, make a cleft in the central region of the protein. As in shown in Figure 2C (bottom), a close-up view with the ball-and-stick modeling indicates that the substitution of the side chain from Tyr at position 50 to Ser opens the ‘lid’ of the cleft and slightly alters the structure around helix H1 and the adjacent H1′-S2 loop. In microtubules, the H1′-S2 loop is located spatially close to the M loop (microtubule loop) of another β-tubulin of the juxtaposed protofilament [46] (Figure 2C and Appendix A).

This indicates that Y50, together with the H1′-S2 loop, may be involved in ββ lateral interaction. It is possible then that the Y50S mutation in Nda3-TB101 might increase the affinity to the M loop of the neighboring β-tubulin, which may strengthen ββ lateral interaction. Such microtubules containing the mutant β-tubulin might endure the mechanical stress that promotes depolymerization, which might be a reason why the *nda3-TB101* mutant resists benzimidazole.

### 3.3. Two Possible Modes for Y50S β-Tubulin

A number of mutations in β-tubulin that confer resistance to benzimidazole compounds have been identified, mostly in pathogenic fungi and parasitic nematodes, as introduced above. It was recently reported that the putative MBC binding site of β-tubulin in *P. xanthii* is S138 and T178, but not E198, although the mutation E198A confers resistance to a number of fungal species [34]. The E198A mutation in *P. xanthii* β-tubulin is thought to cause conformational alteration that confers MBC resistance, and therefore E198 may not be the binding site for MBC.

On the contrary, a structural analysis proposes that the region containing Y50 is predicted to be a binding site for benzimidazole [47], although there is no biological evidence for this. As the *nda3-TB101* (Y50S) mutation conferred TBZ resistance to *S. pombe*, it is possible that Y50 in *S. pombe* β-tubulin is another binding site for benzimidazole or that Y50S mutation deteriorates its affinity to TBZ, even though Y50 is not the direct binding site for benzimidazole.

We thus predicted mainly two possible ways to interpret TBZ resistance of the *nda3-TB101* mutant, although these two are not mutually exclusive: (i) stabilization of microtubules and (ii) reduced binding of TBZ to the mutant β-tubulin.

### 3.4. Microtubules Containing Nda3-TB101 Are Unaffected by TBZ and MBC

We therefore examined the former possibility that microtubules are stabilized by the *nda3-TB101* mutation. To visualize microtubules, the α2-tubulin (Atb2) gene was fused with GFP and the fusion gene was inserted at a chromosome locus to express it under the *atb2* promoter (Z2-GFP-Atb2, see Materials and Methods).

In the absence of microtubule drugs, both WT (*nda3*^+^) and mutant (*nda3-TB101*) cells displayed a normal array of microtubules in the cytoplasm in interphase (Figure 3A). The number and the length of microtubules did not substantially differ in WT and *nda3-TB101* cells (Figure 3B). When TBZ or MBC (50 μg/mL) was added to the culture of WT cells, microtubule bundles in the cytoplasm diminished almost completely, although some remnants remained in the nucleus under these conditions (Figure 3A,B). In the *nda3-TB101* mutant, however, microtubule organization was not affected by the drugs (Figure 3A,B). Similar results had been obtained with antitubulin immunofluorescence images in the previous study in order to demonstrate the performance of MBC in comparison to TBZ [48].

These results demonstrate that microtubules assembled in the *nda3-TB101* mutation appeared normal in the absence of benzimidazole drugs and that microtubules in the mutant are insensitive to the drugs.

### 3.5. The nda3-TB101 Mutant Is Resistant to Nocodazole

Nocodazole, another derivative of benzimidazole compounds, is commonly employed to inhibit microtubule growth in animal cells and budding yeast. Nocodazole has been rarely used in *S. pombe* studies as the drug does not inhibit growth of WT *S. pombe* cells [33].

As demonstrated in the previous study, this is due to the active efflux of nocodazole from *S. pombe* cells [49]. Briefly, seven endogenous genes have been reported as factors conferring the multidrug resistance (MDR) to *S. pombe*, and the drug efflux can be disabled by deletion of the genes. The strain in which the drug efflux pathways are dismantled is called MDR-sup (MDR-suppressed) [42,49].

The MDR-sup strain in the *nda3*^+^ background was indeed sensitive to nocodazole on agar plates as previously demonstrated (Figure 4A). In line with the plate-based assay, microtubule organization in MDR-sup cells was severely impaired by nocodazole, similarly to TBZ or MBC (Figure 4B). We therefore tested whether the *nda3-TB101* mutant exhibits resistance to nocodazole as seen in the cases of TBZ and MBC in the MDR-sup background. The *nda3-TB101* MDR-sup strain grew better than the *nda3*^+^ MDR-sup strain in the presence of nocodazole (Noc 10 µg/mL, Figure 4A). In line with plate-based assays, microtubule organization in *nda3-TB101* MDR-sup was not damaged by nocodazole (Figure 4B). These results together demonstrate that the *nda3-TB101* mutation conferred resistance to nocodazole as well as to TBZ and MBC.

### 3.6. Microtubule Dynamics in nda3-TB101 Were Moderately Stabilized in the Absence of Drugs

We further focused on the dynamics of cytoplasmic microtubules in *nda3-TB101* cells expressing GFP-Atb2 in order to investigate whether the mutation altered the dynamic behavior of microtubules in vivo.

In the absence of microtubule drugs, frequencies of microtubule catastrophe and rescue in *nda3-TB101* cells slightly decreased by ~30% on average compared to those in WT cells (MDR^+^, Figure 5A). This suggests that the *nda3-TB101* mutation (Y50S in β-tubulin) moderately stabilized the dynamics of microtubules even in the absence of microtubule drugs. This tendency was maintained in cells of the MDR-sup genetic background (MDR-sup, Figure 5A): microtubule dynamics were not altered by introducing the MDR-sup background and were moderately stabilized by the *nda3-TB101* mutation, as was also seen in the MDR^+^ background.

We next investigated effects of microtubule poisons on microtubule dynamics of the *nda3-TB101* cells. As shown in Figure 5B,C, microtubule dynamics were not altered by either TBZ, MBC, nor nocodazole: frequencies of catastrophe and rescue did not change in comparison to those without drugs. These results indicate that the Y50S mutation of β-tubulin per se moderately stabilized microtubule dynamics, and the property was not altered by the addition of benzimidazole compounds.

### 3.7. Microtubule Nucleation Was Not Affected by the nda3-TB101 Mutation

As the dynamics of microtubules seemed slightly stabilized in the *nda3-TB101* mutant, we next tested whether nucleation of microtubules is also promoted by the mutation. Microtubule nucleation during interphase occurs in the nuclear periphery toward the cytoplasm [50]. Growing cells expressing GFP-Atb2 in WT and *nda3-TB101* were prepared and microtubule behavior was monitored in live-cell imaging. When a dot signal of GFP-Atb2 emerged in the cytoplasm, we counted it as one event corresponding to microtubule nucleation in a fluorescence microscopy grade. As shown in Figure 6A, the frequency of nucleation events was not significantly promoted in the *nda3-TB101* mutant.

To further investigate this, cold-shock and recovery experiments were performed using WT and *nda3-TB101* cells. *S. pombe* cells tend to nucleate microtubules in the vicinity of the nuclear envelope gradually upon recovery from cold shock [50]. Microtubules in WT and *nda3-TB101* cells were destroyed to a similar degree by cold shock on ice (CS, Figure 6B), and the cultures were shifted back to room temperature to induce microtubule regrowth. Aliquots of the cells were collected every minute for immunofluorescence to monitor microtubule recovery over time, and percentages of cells with regrown microtubules were initially lower in *nda3-TB101* than in WT but became almost comparable in 4–5 min (Figure 6B,C).

These results demonstrate that the *nda3-TB101* mutation did not promote microtubule nucleation.

### 3.8. The nda3-TB101 Mutation Suppressed the TBZ Sensitivity of the alp14 Mutant

As the Y50S mutation of β-tubulin conferred resistance to benzimidazole compounds, it may suppress the sensitivity to those drugs in other microtubule-related mutants. A representative is the mutant of the *alp14* gene, which encodes an orthologue of TOG/Dis1/XMAP215. The *alp14* knock-out (*alp14*∆) strain displayed short and fragile microtubules and was sensitive to TBZ (*alp14*∆ *nda3*^+^, Figure 7A,B) [9,10]. The double mutant *alp14*∆ *nda3-TB101* grew in the presence of TBZ demonstrating that the Y50S mutation of β-tubulin suppressed the TBZ sensitivity of *alp14*∆ (Figure 7A).

The *alp14*∆ mutant also displays temperature-sensitive phenotypes in the absence of drugs [9,10]. As shown in Figure 7A, *alp14*∆ was indeed defective in growth at a high temperature (36 °C). The *nda3-TB101* mutation partially suppressed the growth defects of the *alp14*∆ mutant at 36 °C (*alp14*∆ *nda3-TB101*, Figure 7A). In line with this, cytoplasmic microtubules of the double mutant were still short at 36 °C but overall organization of microtubules appeared moderately ameliorated compared to *alp14*∆ *nda3*^+^ cells (Figure 7B). Introduction of the *nda3-TB101* mutation into *alp14*∆ cells affected neither frequencies of catastrophe and rescue events (Figure 7C) nor rates of polymerization and depolymerization (Figure 7D). These results demonstrate that the β-tubulin Y50S mutation rescued the TBZ sensitivity of *alp14*∆ efficiently but the microtubule polymerization defects only partially.

We next tested whether Mal3 (the fission yeast EB1), another representative MAP, was affected by the *nda3-TB101* mutation. Mal3 is known to show comet-like localization that consists of enrichment at the plus ends of growing microtubules and the lateral surface around the tips [51]. In the *nda3-TB101* mutant, Mal3 localization to growing tips increased twice of that in WT cells (Appendix A). In contrast to the increase of Mal3 localization to microtubules, fluorescence signal of Alp7-3GFP dots (TACC, which binds to Alp14/TOG) on microtubules in *nda3-TB101* cells was comparable to that in WT (Appendix A).

This indicates that the Y50S β-tubulin accumulated specifically in Mal3 at the growing tips, but not other MAPs. It is possible that microtubules are strengthened by the increased Mal3 localization. The *mal3*∆ cells exhibited short cytoplasmic microtubules around the nucleus [52], and the microtubule length in *mal3*∆ *nda3-TB101* double mutant cells was almost comparable (Appendix A). The mutant β-tubulin protein Nda3-TB101 may not exert its influence on the stability of plus tips of microtubules when Mal3 is absent. It is alternatively possible that the defect of *mal3*∆ was so severe that it simply overwhelmed tightened ββ-contacts of mutant proteins. Another phenotype, TBZ sensitivity, of the *mal3*∆ mutant was dramatically suppressed by introduction of the *nda3-TB101* mutation (Appendix A).

### 3.9. Phenotypic Differences in klp5/6 and nda3-TB101 Mutants

In general, loss-of-function or hypomorphic mutants of tubulin and MAPs show TBZ sensitivity as those factors are required for assembly or stabilization of microtubules. Mutants of kinesin-8 are exceptions: *klp5*∆ and *klp6*∆ mutants are resistant to TBZ [15,16]. This has been interpreted that these kinesin-8s promote depolymerization of cytoplasmic microtubules. We then sought the genetic relationship between the *nda3-TB101* and *klp6*∆ mutants.

Regarding dynamics in the absence of drugs, cytoplasmic microtubules in *klp6*∆ cells decreased frequencies of catastrophe and rescue events to ~70% of WT (Figure 5A and Figure 8A), suggesting that microtubules in both *klp6*∆ and *nda3-TB101* cells were less dynamic than in WT cells. The reduced dynamics in *klp6*∆ were not enhanced by introducing the *nda3-TB101* mutation (Figure 8A), indicating that the Y50S mutation may not be able to exert its influence on microtubules in the absence of kinesin-8.

Regarding drug effects, both *klp6*∆ and *nda3-TB101* are known to be resistant to TBZ. The *klp6*∆ mutant was indeed resistant to 30 µg/mL TBZ on agar plates, but was unable to form colonies in the presence of 75 µg/mL TBZ (Figure 8B). In contrast, the *nda3-TB101* mutant formed colonies in 75–100 µg/mL TBZ, although the viability decreased in accordance with an increase in TBZ dosage (Figure 8B). Notably, growth of the *nda3-TB101 klp6*∆ double mutant was not affected even by 100 µg/mL TBZ, indicating that the phenotype of TBZ resistance was additive (Figure 8B).

When microtubules were visualized with GFP-Atb2 in the absence of the drug, both *klp6*∆ and *nda3-TB101 klp6*∆ cells displayed bundles of cytoplasmic microtubules, the ends of which were frequently curved (TBZ 0, Figure 8C), as was previously characterized for *klp6*∆ cells [15,16]. It is notable, however, that the curvy array of microtubules was not evident in *nda3-TB101* mutant cells, indicating that microtubules were not so stabilized as in *klp6*∆ cells (Figure 3A). Microtubules in the *klp6*∆ and *nda3-TB101* mutants may be stabilized in a distinct manner. In *nda3-TB101 klp6*∆ cells, the curvy array of stabilized microtubules tolerated higher doses of TBZ (50–100 µg/mL), in line with their steady growth in high doses of TBZ (Figure 8C).

These results led us to postulate a scenario how microtubules are stabilized in these mutants. First, microtubules in *klp6*∆ are stabilized owing to the loss of microtubule depolymerizing factors, therefore these microtubules tolerate microtubule poisons when exposed to low doses. On the contrary, microtubules in *nda3-TB101* are stabilized only moderately in the absence of the poisons, possibly through augmented ββ-interaction or by accumulated Mal3. Besides that, microtubules in *nda3-TB101* appear remarkably insensitive to the poisons, even when exposed at high doses.

### 3.10. The Y50S Mutation of β-Tubulin Interferes Binding of TBZ

It is therefore possible that Y50S substitution in β-tubulin causes it to lose its affinity to benzimidazole drugs. This possibility was tested in vitro affinity assays using recombinant GST-fused β-tubulin proteins. Recombinant proteins (GST-Nda3 (WT), GST-Nda3-TB101 and GST) expressed in bacterial cells were immobilized in sepharose columns (Figure 9A,B).

TBZ solution was then applied to each column and then solutions flowing through the columns were sequentially collected into fractions over time. Absorbance at 297 nm for each solution was measured to detect TBZ [43] and eluted fractions (numbers 6–9) collected from the Nda3-TB101 column showed higher absorbance than those from the Nda3 (WT) column (Figure 9C). The flow through of TBZ from the control GST column was relatively constant from the first fraction until late ones. The immediate flow through of TBZ indicates that GST in the control column did not trap TBZ from the beginning. In contrast, TBZ flow through was at first low in GST-Nda3 and GST-Nda3-TB101 columns, probably because both proteins at first trapped TBZ to a certain degree, but GST-Nda3-TB101 subsequently failed to retain the bound TBZ to result in the highest elution of TBZ in late fractions. In summary, comparison of results from GST-Nda3 and GST-Nda3-TB101 columns demonstrates that TBZ has a higher affinity to the wild-type Nda3 than to Nda3-TB101 in vitro.

## 4. Discussion

This study first reconfirmed that the β-tubulin mutant *nda3-TB101* confers resistance to benzimidazole compounds such as TBZ and MBC, as reported in 1980 [32]. The mutant also exhibited resistance to nocodazole in the MDR-sup genetic background. The mutation is a substitution of tyrosine at the position 50 with serine. In this study, we also investigated the mechanism of how the mutant is resistant to the drugs, employing recent advancements in genetics and microscopy.

Our observation indicated that the Y50S (TB101) mutation provides two possible effects on the nature of β-tubulin and microtubules. These can be discussed depending on two situations: in the presence or absence of benzimidazole compounds.

First, in the absence of drugs, Y50S β-tubulin may augment microtubule robustness. The H1′-S2 loop of β-tubulin in the N-terminal region including Y50 is predicted to the region for lateral contact to a juxtaposed β-tubulin in microtubules, assuming that the major pattern of lateral contact in vivo is αα as well as ββ, called the B-lattice [53] (Appendix A). It is therefore possible that the Y50S mutation stabilizes ββ interactions in microtubules that might allow robustness against catastrophe or depolymerization even in the absence of microtubule poisons. The reduction of the catastrophe frequency may be ascribed to the stabilization of lateral contacts by Y50S (see Figure 5).

As Mal3 (EB1) localization moderately increased at the plus ends as well as the lateral surface of microtubules (Appendix A), it is possible that the Y50S mutation affected binding of Mal3 to microtubule dimers. The Mal3 binding site in the microtubule lattice appears distal from Y50 (Appendix A), therefore its accumulation can be interpreted either as a secondary effect through the stabilization of ββ lateral contact or through allosteric conformational change by Y50S substitution around the Mal3 binding site.

In relation to the latter possibility, the GTP binding site may be affected by Y50S substitution. Structural and biochemical analyses have proven that residues 12, 63–77 are located by the guanine base of GTP, 155–174 are by ribose and 103–109 are thought to affect the GTP hydrolysis [1,54]. Notably, Y50 is located alongside those GTP-binding regions (Appendix A) and therefore it is possible that Y50S substitution somehow negatively affected GTP hydrolysis so that the GTP-bound state can be prolonged in microtubules comprising Y50S β-tubulin. EB1 recognizes the nucleotide state of tubulin in the plus tip and the lattice of microtubules and preferentially binds to the GTP-bound state [5,6,55,56], therefore it is conceivable that the Y50S substitution increased Mal3 loading as a result of possible conformational change around the GTP-binding region.

Thus, phenotypes observed in the absence of microtubule poisons may be interpreted in either way: stabilized ββ lateral contact, increased GTP-bound β-tubulin, and/or increased EB1 along microtubules. Further investigation will be required to test these hypotheses. In any case, such stabilized microtubules in *nda3-TB101* did not appear to bring detrimental effects to mitotic cycles and to sexual differentiation (cell conjugation and meiosis; data not shown).

Second, Y50S β-tubulin may exert its power in the presence of drugs. In addition to augmentation of the microtubule stability, microtubules containing Y50S β-tubulin displayed remarkable tolerance to high doses of benzimidazole. The Y50S mutation of the β-tubulin lost its affinity to TBZ (Figure 9) and this is the main reason for the TBZ resistance.

Discussion regarding the binding site of benzimidazole to β-tubulin is still controversial. In general, experimental data from chemical genetics have been used to predict the binding site: namely, the site of the resistant mutation is a major candidate for the binding site. Mutations close to position Y50 were previously isolated for the β-tubulin (Tub2) of the budding yeast *Saccharomyces cerevisiae.* A systematic alanine scanning was performed [25] and *S. cerevisiae tub2-410* (D40A, D41A) and *tub2-411* (K44A, E45A, R46A) mutants were found resistant to benomyl, albeit only moderately. The minor phenotype in the mutants may be attributed to a change in microtubule dynamics rather than a change in drug binding, as discussed previously [57]. The alanine scanning also identified mutations in the region 194–198 that conferred stronger resistance, therefore those residues, rather than 40–46, may be engaged in benomyl binding at least for *S. cerevisiae* cells.

In a previous study, information of mutation sites that conferred benzimidazole resistance was employed as input for computational modeling. The model estimated that the binding site of albendazole sulphoxide (ABZ-SO) in *H. contortus* β-tubulin lies at the interface between the N-terminal and the intermediate domains. Namely, ABZ-SO was deduced to bind to residues H6, Y50, Q134, S165 (A165 in *S. pombe*), F167, E198, F200, and M257 [47] (Appendix A). This region also includes E198, G235, and C239 (S239 in *S. pombe*), which are essential to bring benzimidazole sensitivity in the bird *Gallus gallus* [58].

In contrast, an *S. pombe* mutant of β-tubulin E198A was recently created from analogy to the *P. xanthii* mutant with the same mutation [34]. Although E198A mutants of both fungal species were resistant to MBC, their computational operations pinpointed regions containing S138 and T178 are putative but the best-fit docking sites for MBC (Appendix A). In that study, technical limitations hampered chemical assays testing whether mutations in S138 or T178 lose affinity to MBC, therefore it remains elusive whether those residues are the binding site.

Thus, these studies proposed mainly two distinct regions for benzimidazole binding. The first candidate region, estimated as ABZ-SO binding region, does not overlap with but is juxtaposed to the second region, estimated as MBC binding site containing S138 and T178 of *P. xanthii* β-tubulin (Appendix A). It is notable that the first region includes E198, which was excluded from the second candidate region, although the E198A mutation conferred resistance to MBC in the study [34].

Our results provide genetic and cytological evidence for those computational models proposed in the past. The Y50S mutation in *S. pombe* β-tubulin significantly reduced affinity to TBZ and the mutant exhibited remarkably strong resistance to benzimidazole, indicating that Y50 may be directly involved in binding to benzimidazole rather than just being a piece to support conformation of β-tubulin. Y50 is located at the edge of the first candidate region but is on the opposite side of the second region (Appendix A). These results thus provide genetic evidence to support the first model: the region including H6, Y50, Q134, A165, F167, E198, Y200, and M257 may be the binding site for benzimidazole. Y50 possibly serves as a wall of the benzimidazole-binding pocket and Y50S substitution may cause a loss of the wall in the pocket so that benzimidazole could easily dissociate therefrom (Appendix A).

Taking all results together, we propose that the region may be the binding site of benzimidazole in *S. pombe* β-tubulin. Further structural analyses of Y50S β-tubulin and benzimidazole are required to test the idea. Our observations demonstrate that the *nda3-TB101* mutation has a dual impact on the nature of β-tubulin and microtubules: (1) microtubules containing the Y50S mutant β-tubulin are stabilized against internal depolymerization cues and (2) the mutant β-tubulin Y50S is insensitive to benzimidazole-type microtubule drugs.

Investigation of resistant strains through a combination of structural and genetic assays will give us pharmacological insights into the mechanical actions of the benzimidazole compounds with tubulins as the target. This may lead to drug designs for novel benzimidazole derivatives that work against emerging resistant strains of pathogenic or parasitic organisms.

## Figures and Tables

**Figure 1 cells-10-01042-f001:**
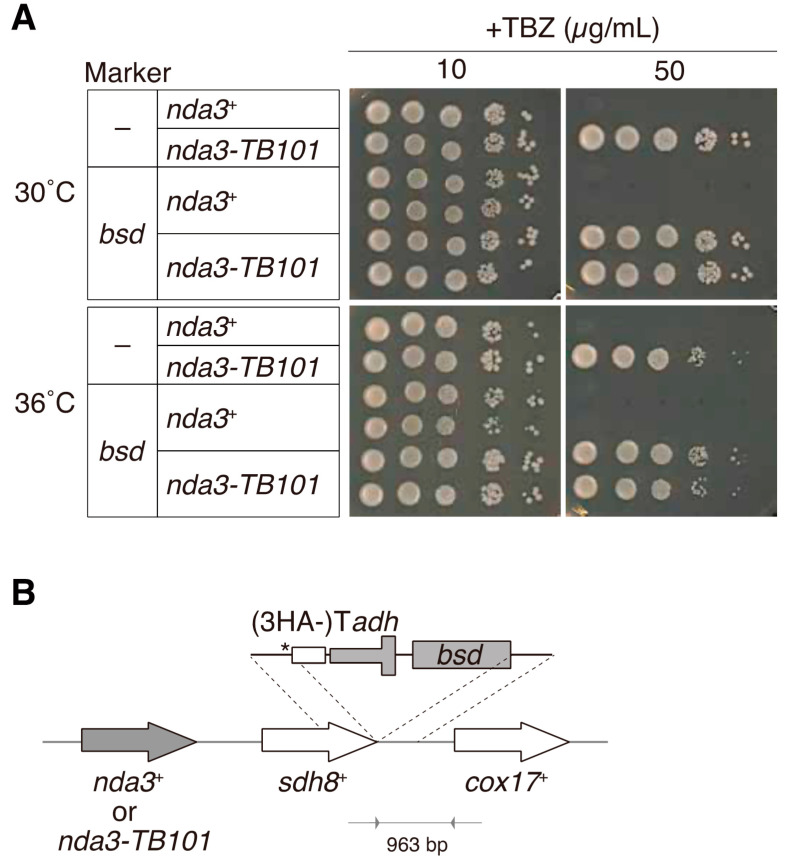
The β-tubulin mutant *nda3-TB101* is resistant to TBZ. (**A**) The indicated strains were tested for TBZ resistance to TBZ. The *nda3*^+^ has the WT *nda3* gene. The column “marker” indicates whether the blasticidin S resistance gene (*bsd*) was inserted near the *nda3* locus as illustrated in B, not inserted (–). Ten-fold serial dilutions of cells with indicated genotypes were spotted on the rich medium containing TBZ (10 or 50 µg/mL) and the plates were incubated at 30 °C or 36 °C. (**B**) A schematic illustrating the construct of marker insertion into a chromosome. The *bsd* gene was inserted at the downstream of the *sdh8*^+^ gene, which located next to the *nda3*^+^ (or *nda3-TB101*) gene. To guarantee the expression of the *sdh8*^+^ gene, the *adh1* terminator (Tadh) was inserted followed by *bsd*. The DNA sequence encoding three copies of HA (3HA) was also inserted at the 3’ end of *sdh8*^+^, but the termination codon (asterisk) was placed to avoid fusion of Sdh8 protein with 3HA.

**Figure 2 cells-10-01042-f002:**
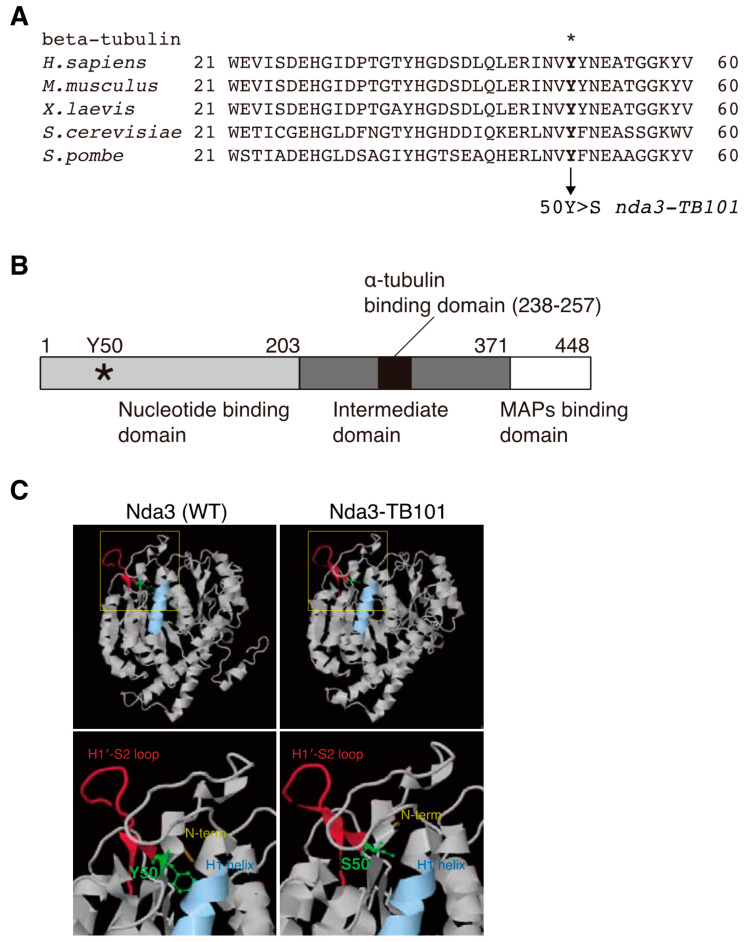
The Y50S mutation is found in the mutant β-tubulin Nda3-TB101. (**A**) Alignment of a part of β-tubulin proteins (aa 21–60) of indicated organisms. The *S. pombe* β-tubulin mutant protein Nda3-TB101 contains a single missense mutation that substitutes the conserved tyrosine residue (asterisk) to a serine (Y50S). (**B**) A domain structure of *S. pombe* β-tubulin Nda3. Y50 (asterisk) resides in the nucleotide (GTP/GDP) binding domain in the N-terminus. (**C**) Predicted structures of Nda3 (WT) and Nda3-TB101 containing Y50S substitution. The Phyre2 algorithm was employed to predict protein structures. The entire structure (top). Insets include the Y50 (or S50) containing regions and are shown magnified (bottom). Red, H1′-S2 loop; green, Y50 and S50; yellow, N-terminal end; light blue, H1 helix.

**Figure 3 cells-10-01042-f003:**
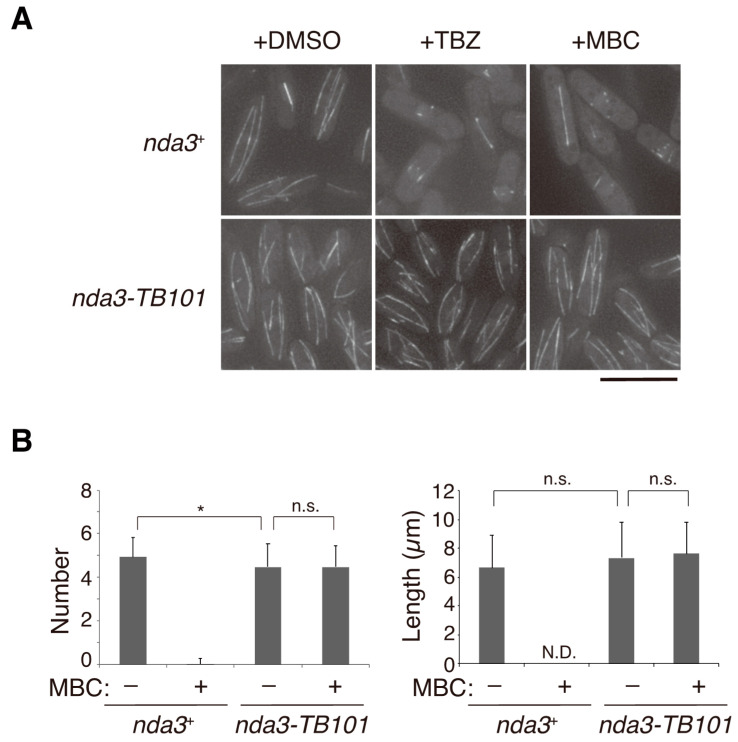
Microtubules in *nda3-TB101* cells are unaffected by microtubule poisons. (**A**) WT (*nda3*^+^) and *nda3-TB101* cells expressing GFP-Atb2 (α2-tubulin) were grown and observed in the presence of DMSO (control), TBZ, and MBC (50 µg/mL each). Scale bar, 10 µm. (**B**) The numbers of cytoplasmic microtubules per cell (left, *n* = 50 cells) and the length of each microtubule bundle (right, *n* = 50 bundles) in WT and *nda3-TB101* cells were measured in the absence or presence of MBC (50 µg/mL). N.D., not determined; n.s., not significant; * *p* = 0.021 < 0.05 (Student’s *t*-test); error bars = S.D.

**Figure 4 cells-10-01042-f004:**
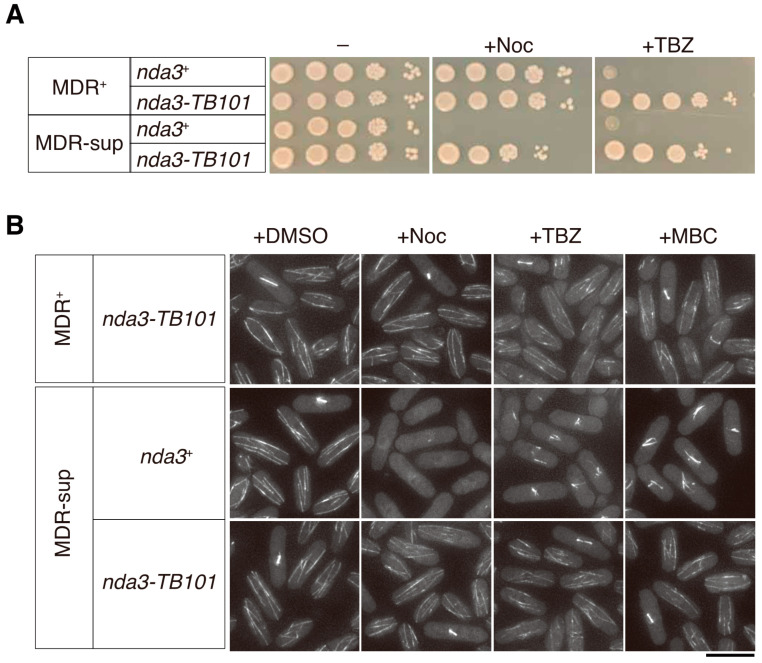
The *nda3-TB101* mutation confers resistance to nocodazole in the drug-vulnerable strain. (**A**) The *nda3-TB101* strain in the MDR-sup (multi-drug resistance suppressed) background was used to test sensitivity to nocodazole (Noc, 10 µg/mL) and TBZ (30 µg/mL). MDR^+^, standard genetic background. Ten-fold serial dilutions of indicated strains were spotted on rich media without (–) or with nocodazole or TBZ. The plates were incubated at 30 °C for 4 days. (**B**) Indicated strains expressing GFP-Atb2 were grown at 30 °C in the absence of drugs (+DMSO) as well as in the presence of nocodazole (+Noc, 10 µg/mL), TBZ (50 µg/mL), and MBC (50 µg/mL). Scale bar, 10 µm.

**Figure 5 cells-10-01042-f005:**
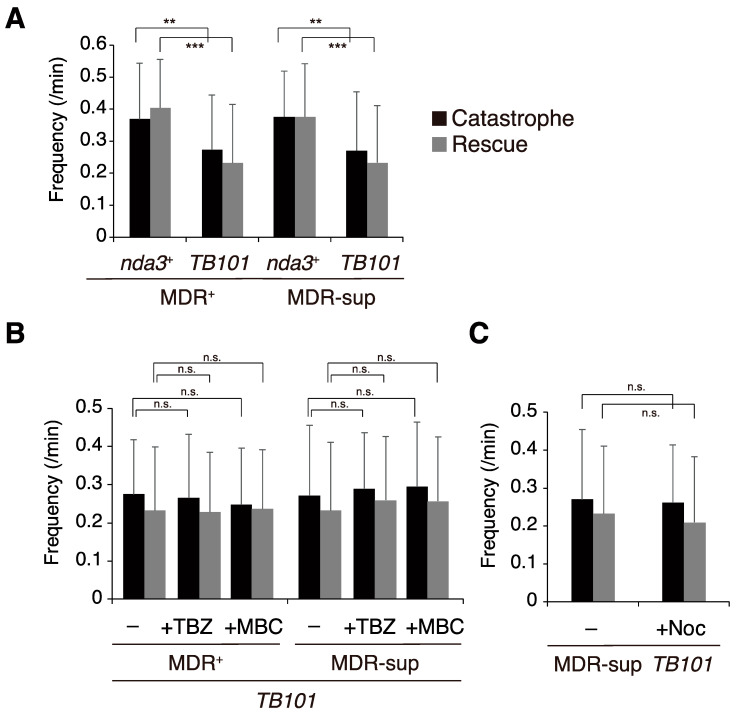
Microtubules in *nda3-TB101* cells are less dynamic than those in WT cells. (**A**) Frequencies of catastrophe and rescue in cytoplasmic microtubules were monitored in the absence of drugs using WT (*nda3*^+^) and *nda3-TB101* strains expressing GFP-Atb2 in the MDR^+^ (left) and MDR-sup (right) backgrounds. In both backgrounds, microtubules in *nda3-TB101* cells reduced frequencies of both catastrophe and rescue. MDR^+^
*nda3*^+^, *n* = 56 bundles; MDR^+^
*nda3-TB101*, *n* = 56; MDR-sup *nda3*^+^, *n* = 54; MDR-sup *nda3-TB101, n* = 53. Left, ** *p* = 0.0022 < 0.01; *** *p* = 9.4 × 10^−8^ < 0.001 (Student’s *t*-test). Right, ** *p* = 0.0027 < 0.01; *** *p* = 8.5 × 10^−5^ < 0.001 (Student’s *t*-test). (**B**) Frequencies of microtubule catastrophe and rescue in the *nda3-TB101* mutant in the presence of TBZ (50 µg/mL), MBC (50 µg/mL), or DMSO (for control, –) were monitored as in A. Left (MDR^+^), +DMSO, 56 bundles; +TBZ, *n* = 54; +MBC, *n* = 55. Right (MDR-sup), +DMSO, *n* = 53 bundles; +TBZ, *n* = 54; +MBC, *n* = 52. Not significant, n.s. (Student’s *t*-test). (**C**) Frequencies of microtubule catastrophe and rescue were monitored in the presence of nocodazole (15 µM) or DMSO (for control, –). +DMSO, *n* = 51 bundles; +Noc, *n* = 53. Note that samples for control (–, MDR-sup *TB101*) are shared in experiments for **B** and **C**.

**Figure 6 cells-10-01042-f006:**
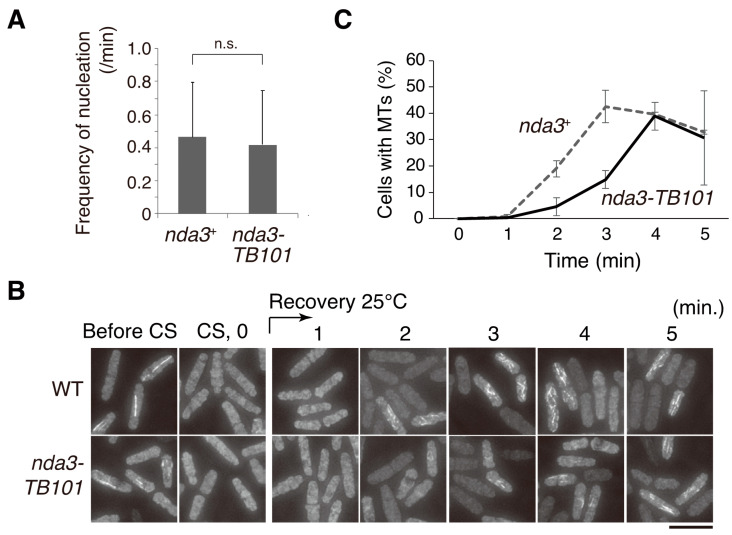
Microtubule nucleation is not promoted by the *nda3-TB101* mutation. (**A**) Frequency of microtubule nucleation events was monitored during interphase of WT (*nda3*^+^) and *nda3-TB101* cells expressing GFP-Atb2. When a punctate signal of GFP-Atb2 appeared in the cytoplasm and then grew to a filament, the event was counted as microtubule nucleation in this study. The frequency of nucleation per minute per cell was counted. Cells, *n* = 20. Not significant, n.s. (Student’s *t*-test). (**B**,**C**) Microtubules were depolymerized by cold-shock treatment in WT and *nda3-TB101* cells expressing GFP-Atb2, and regrowth was induced by temperature shift back. Microtubules were observed before (Before CS) and after (CS, 0 min) giving a cold shock (**B**). Microtubule regrowth was monitored every minute after the temperature shifted back. Scale bar, 10 µm. Percentages of cells harboring punctate and/or filament signal of GFP-Atb2 at each time point (**C**). Cells, n > 300.

**Figure 7 cells-10-01042-f007:**
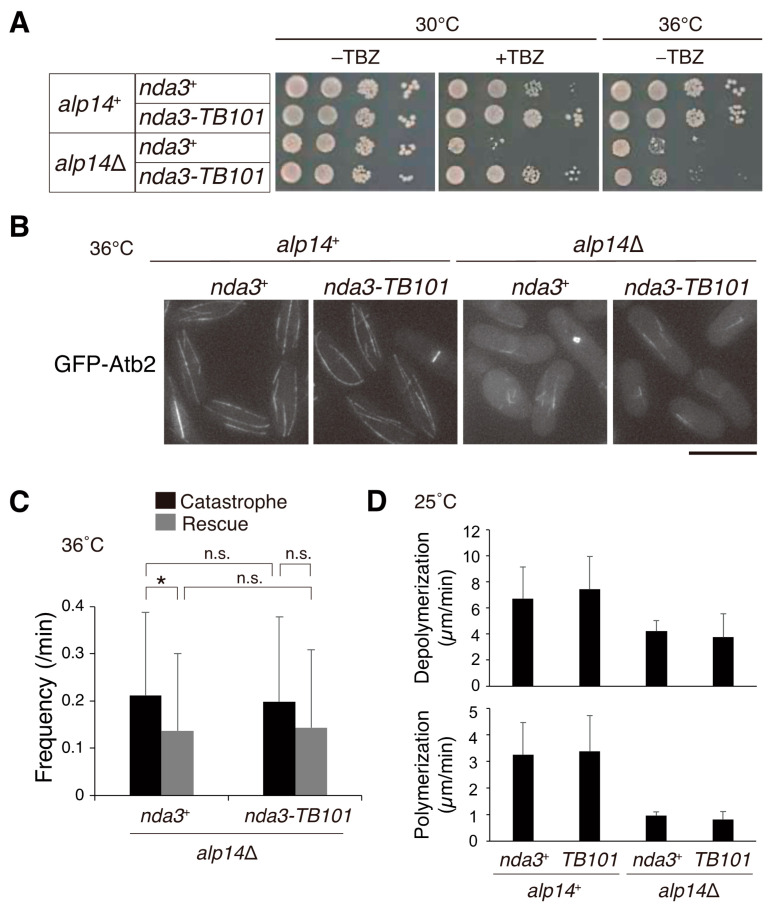
The *nda3-TB101* mutation partially suppresses phenotypes of the *alp14* deletion mutant. (**A**) Ten-fold serial dilutions of cells with indicated genotypes in rich media without (–TBZ) or with TBZ (10 µg/mL). Plates were incubated at 30 °C or 36 °C for 4 days. (**B**) Behavior of cytoplasmic microtubules at 36 °C was monitored with GFP-Atb2 in cells with indicated genotypes. Scale bar, 10 µm. (**C**) Frequencies of microtubule catastrophe and rescue at 36 °C per minute per cell were counted. Bundles, n > 40. * *p* = 0.041 < 0.05; n.s., not significant. (**D**) Velocities of microtubule polymerization and depolymerization were calculated in cells with the indicated genotypes (25 °C). Bundles, *n* = 10.

**Figure 8 cells-10-01042-f008:**
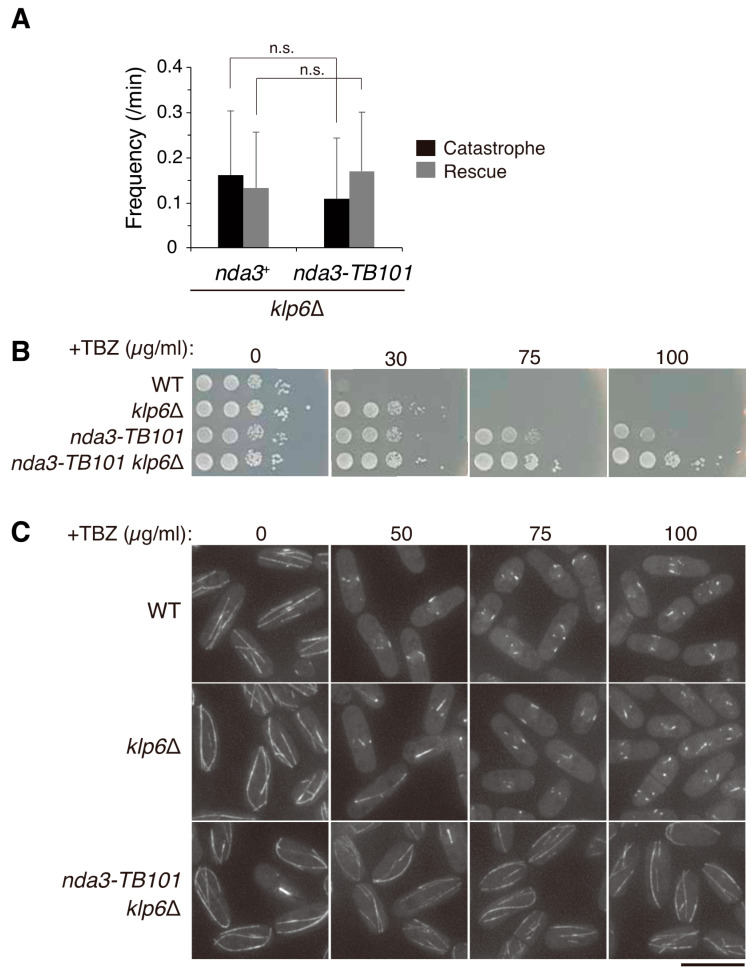
Microtubules are differently stabilized in the mutants of kinesin-8 and *nda3-TB101*. (**A**) Frequencies of catastrophe and rescue of microtubules in the indicated strains. The *nda3*^+^ gene (WT), *n* = 21 bundles; *nda3-TB101* gene, *n* = 20. Not significant, n.s. (Student’s *t*-test). (**B**) Ten-fold serial dilutions of cells with indicated genotypes in rich media without (0) or with TBZ at indicated concentrations (30–100 µg/mL). Incubated at 30 °C for 3 days. (**C**) Microtubules were monitored in cells expressing GFP-Atb2 in the indicated genotypes at each concentration of TBZ (0–100 µg/mL). Scale bar, 10 µm.

**Figure 9 cells-10-01042-f009:**
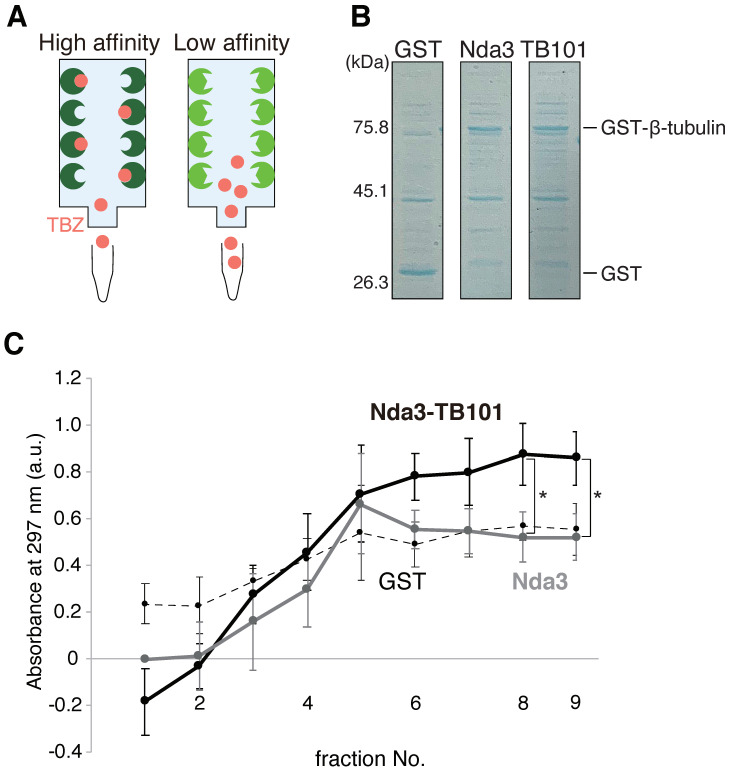
The Y50S mutation of β-tubulin reduces binding affinity to TBZ. (**A**) A diagram for the binding assays. Recombinant β-tubulin proteins are immobilized in the column and a TBZ solution is applied. TBZ is to be efficiently captured in the column when the affinity between β-tubulin and TBZ is high whilst TBZ will flow through the column when the affinity is low. (**B**) Expression of recombinant proteins Nda3 (WT) and Nda3-TB101 was confirmed by SDS-PAGE followed by Coomassie staining. MW (left) is judged from a marker. (**C**) Sequentially collected fractions 1–9 were applied to measure absorbance at 297 nm (a.u.). Gray line, Nda3 (WT); black solid line, Nda3-TB101; dashed line, GST. Error bars, S.D., * *p* < 0.05 (0.043, fraction 8; 0.035, fraction 9) (Student’s *t*-test), *n* = 3 experiments.

## Data Availability

The data presented in this study are available from the corresponding author upon reasonable request.

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
