# Peer review of "Dual Impact of a Benzimidazole Resistant β-Tubulin on Microtubule Behavior in Fission Yeast"

_cells, 2021, doi:10.3390/cells10051042_

Round 1

Reviewer 1 Report

This manuscript by Minagawa et al. characterizes the S. pombe β-tubulin mutant nda3-TB101, which was previously isolated as a mutant resistance to benzimidazole. First, they identified the mutation site tyrosine at position 50. Second, they found two properties of the mutant tubulin. The dynamics of cellular microtubules comprising the mutant β-tubulin was stabilized in the absence of benzimidazole. In addition, the mutant protein reduced the affinity to benzimidazole in vitro. The experiments are well designed, well executed and the data is convincing. I think that this work should be published, but before publication one negative control experiment described below need to be addressed. 1) In Figure 9C in vitro experiments, GST alone control would be a good negative control to evaluate how the Y50S mutation loses affinity to benzimidazole.

Author Response

Our reply:

We thank the reviewer for the positive comments.

We consider that comparison of GST-Nda3 (WT) and GST-Nda3-TB101 would be easier and more simple for readers to directly tell the difference of WT Nda3 and Nda3-TB101 in TBZ affinity. Although interpretation for the GST column is a bit confusing, but is now explained in the text (lines 566-573), to sincerely address the issue raised by the reviewer. We thank for the reviewer for the suggestion.

Accordingly, the method section was also edited (Section 2.7).

Reviewer 2 Report

The authors analyse the dual impact of the benzimidazole resistant nda3-TB101 (Y50S) mutant in fission yeast. They confirm the sensitivity and analyse microtubules dynamics. Finally, they combine the mutant with other well-characterized mutation in alp14, mal3 and klp6 and they perform an invitro column retention assay of TBZ.
Altogether, the experiments are properly performed and controled. The conclusion that the mutation stabilizes the microtubules even in the absence of the drug is of interest for the cytoskeletton community. 

Comments:
- the authors report the insertion of the bsdR gene downstream of sdh8 but don’t tell why they did this experiment. As this is presented early in the manuscript, the reader expect that thuis marker will be usefull but this completely disappear during the rest of the paper. This should be explained and the paragrap may be moved to the material and methods.

Figure 1
It is hard to read the figure. The horizontal lines separating the names should continue on the plates or be placed on the left side. Indeed, the fact that some strain do not grow on the righ panel makes it difficult to read the name of teh strain.

Figure 8B
TB101 should be written full: nda3-TB101

- It would be usefull to test the fertility of the strain. Indeed, some nda3 mutants (including the F200Y) have meiotic specific defects while happy during vegetative growth. It would be interested to test this by checking an h90 strain on malt extract and iodine staining, or more quantitatively by counting tetrad and spore viability. Indeed the stabilized microtubules may be detrimental during meiosis.

Author Response

The authors analyse the dual impact of the benzimidazole resistant nda3-TB101 (Y50S) mutant in fission yeast. They confirm the sensitivity and analyse microtubules dynamics. Finally, they combine the mutant with other well-characterized mutation in alp14, mal3 and klp6 and they perform an invitro column retention assay of TBZ.
Altogether, the experiments are properly performed and controled. The conclusion that the mutation stabilizes the microtubules even in the absence of the drug is of interest for the cytoskeletton community. Comments:
- the authors report the insertion of the bsdR gene downstream of sdh8 but don’t tell why they did this experiment. As this is presented early in the manuscript, the reader expect that thuis marker will be usefull but this completely disappear during the rest of the paper. This should be explained and the paragrap may be moved to the material and methods. Our reply: We now mention the reason why the bsd marker was inserted near the nda3-TB101 locus in lines 242-243 in results (Section 3.1). In addition, we now mention in which experiments the bsd-marked strains were used, in the method section (lines 142-145, Section 2.2 and Supplementary table S1).
Figure 1
It is hard to read the figure. The horizontal lines separating the names should continue on the plates or be placed on the left side. Indeed, the fact that some strain do not grow on the righ panel makes it difficult to read the name of teh strain. Our reply: According to the suggestion, the strain names are now placed on the left side in Figure 1.
Figure 8B
TB101 should be written full: nda3-TB101 Our reply: According to the indication, the genotype is now corrected as nda3-TB101 in Figure 8B.
- It would be usefull to test the fertility of the strain. Indeed, some nda3 mutants (including the F200Y) have meiotic specific defects while happy during vegetative growth. It would be interested to test this by checking an h90 strain on malt extract and iodine staining, or more quantitatively by counting tetrad and spore viability. Indeed the stabilized microtubules may be detrimental during meiosis. Our reply: We have briefly tested this possibility, and found that the nda3-TB101 mutant underwent normal mating and meiosis to produce viable spores. The data was not shown because of the time limitation, but briefly mentioned in the Discussion part of the text (lines 621-623) for information. We thank the reviewer for constructive comment.